# GTS-21 Enhances Regulatory T Cell Development from T Cell Receptor-Activated Human CD4^+^ T Cells Exhibiting Varied Levels of *CHRNA7* and *CHRFAM7A* Expression

**DOI:** 10.3390/ijms241512257

**Published:** 2023-07-31

**Authors:** Masato Mashimo, Takeshi Fujii, Shiro Ono, Yasuhiro Moriwaki, Hidemi Misawa, Tetsushi Azami, Tadashi Kasahara, Koichiro Kawashima

**Affiliations:** 1Department of Pharmacology, Faculty of Pharmaceutical Sciences, Doshisha Women’s College of Liberal Arts, Kyotanabe 610-0395, Japan; mmashimo@dwc.doshisha.ac.jp (M.M.); tfujii@dwc.doshisha.ac.jp (T.F.); 2Laboratory of Immunology, Faculty of Pharmacy, Osaka Ohtani University, Tondabayashi 584-8540, Japan; onos@osaka-ohtani.ac.jp; 3Department of Pharmacology, Faculty of Pharmacy, Keio University, Minato-ku, Tokyo 105-8512, Japan; moriwaki-ys@pha.keio.ac.jp (Y.M.); misawa-hd@pha.keio.ac.jp (H.M.); 4Division of Gastroenterology, Department of Internal Medicine, Showa University Fujigaoka Hospital, Yokohama 227-8502, Japan; azamitetsushidesu@gmail.com; 5Division of Inflammation Research, Jichi Medical University, Shimotsukeshi 324-0498, Japan; kasahara-td@oregano.ocn.ne.jp; 6Department of Molecular Pharmacology, Kitasato University School of Pharmaceutical Sciences, Minato-ku, Tokyo 108-8641, Japan

**Keywords:** acetylcholine, adoptive immunotherapy, α7, dupα7, GTS-21, nAChR, Treg

## Abstract

Immune cells such as T cells and macrophages express α7 nicotinic acetylcholine receptors (α7 nAChRs), which contribute to the regulation of immune and inflammatory responses. Earlier findings suggest α7 nAChR activation promotes the development of regulatory T cells (Tregs) in mice. Using human CD4^+^ T cells, we investigated the mRNA expression of the α7 subunit and the human-specific dupα7 nAChR subunit, which functions as a dominant-negative regulator of ion channel function, under resting conditions and T cell receptor (TCR)-activation. We then explored the effects of the selective α7 nAChR agonist GTS-21 on proliferation of TCR-activated T cells and Treg development. Varied levels of mRNA for both the α7 and dupα7 nAChR subunits were detected in resting human CD4^+^ T cells. mRNA expression of the α7 nAChR subunit was profoundly suppressed on days 4 and 7 of TCR-activation as compared to day 1, whereas mRNA expression of the dupα7 nAChR subunit remained nearly constant. GTS-21 did not alter CD4^+^ T cell proliferation but significantly promoted Treg development. These results suggest the potential ex vivo utility of GTS-21 for preparing Tregs for adoptive immunotherapy, even with high expression of the dupα7 subunit.

## 1. Introduction

Immune cells such as T cells, B cells and monocytes express various subtypes of both muscarinic and nicotinic acetylcholine (ACh) receptors (mAChRs and nAChRs, respectively) [1,2,3,4,5]. Moreover, they also express mRNA for ACh synthase (choline acetyltransferase, ChAT) and synthesize ACh [1,2,3,4,5,6,7,8,9,10,11]. Activation of T-cell receptors (TCRs) by phytophemagglutinin (PHA) or anti-CD3/CD28 monoclonal antibodies or by activation of protein kinases A and C, enhances ChAT mRNA expression and increases ACh synthesis and release [8,12]. These findings indicate that ACh synthesized by immune cells acts in an autocrine and paracrine manner via AChRs on immune cells, especially T cells, and is involved in regulating immune function.

Among the various mAChRs and nAChRs subtypes expressed by immune cells, α7 nAChR has received much attention. This is because α7 nAChR activation in lipopolysaccharide (LPS)-treated mice prevents septic shock by inhibiting the synthesis and release of the pro-inflammatory cytokine tumor necrosis factor-α (TNF-α) from macrophages [13]. In addition, α7 nAChR gene-deficient (α7-KO) mice immunized with ovalbumin had higher serum concentrations of anti-ovalbumin-specific IgG_1_ than identically treated wild-type mice. At the same time, synthesis of the pro-inflammatory cytokines TNF-α, interferon-γ (IFN-γ) and IL-6 was up-regulated in spleen cells from ovalbumin-immunized α7-KO mice [14]. On the other hand, ACh produced in T cells and α7 nAChRs expressed on macrophages are known to play key roles in various cholinergic anti-inflammatory pathways [15,16,17,18,19,20]. Together, these findings indicate the involvement of α7 nAChRs on immune cells in the regulation of inflammatory and immune functions.

Recently, Mashimo et al. (2019) found that GTS-21, a partial α7 nAChR agonist [21], promotes the differentiation of TCR-activated mouse CD4^+^ T cells into regulatory T cells (Tregs) and effector helper T (Th1, Th2, and Th17) cells [22]. However, GTS-21 suppresses antigen-processing and antigen-presenting cell (APC)-dependent activation of mouse CD4^+^ T cell differentiation. This suggests that α7 nAChRs play a variety of roles affecting immune function, and that their effects depending on the cells in which they are expressed (CD4^+^ T cells or APCs) [22,23].

The gene encoding the human neuronal α7 nAChR subunit *CHRNA7* is located on chromosome 15 and contains 10 exons. Exons 1–6 encode the extracellular N-terminal region of the receptor, including the ligand-binding domain, and exons 7–10 encode the channel region [24]. Notably, chromosome 15 also contains a human-specific partial duplicate α7 nAChR subunit-like gene with exons 5–10. This gene rearranges with the kinase gene *FAM7A* on chromosome 3 to form a hybrid, *CHRFAM7A* [25]. The *CHRFAM7A* gene product, dupα7, lacks a ligand-binding region and assembles with intact α7 subunits to form α7 nAChRs composed of a total of five α7 and dupα7 subunits in various ratios [26]. Because dupα7 acts as a dominant negative regulator of ion channel function, α7 nAChRs with a large dupα7 component do not function well as ion channels, despite retaining of channel structure [26,27,28,29,30,31]. Neuronally expressed α7 nAChRs, which function mainly as ligand-gated ion channels play key roles in cellular signaling, and their dysfunction due to widespread expression of dupα7 is thought to be associated with several central nervous system disorders, including schizophrenia and certain forms of cognitive deficits (see reviews by Bertrand et al. (2015) and Bertrand and Terry (2017)) [32,33]. Multiple clinical trials with α7 nAChR agonists failed to demonstrate efficacy in patients with cognitive deficits or schizophrenia, suggesting decreased expression of functional α7 nAChRs in these patients’ neuronal cells [32,33]. However, most α7 nAChR agonists, including GTS-21, have been shown clinically to be safe [32,34].

Evidence now suggests that α7 nAChRs have dual functions as canonical ionotropic channels and as non-canonical metabolic signaling receptors in both neuronal and non-neuronal cells [35]. α7 nAChRs with metabotropic function are coupled to heterotrimeric G proteins such as Gαq and activate a cascade of signals leading to the release of Ca^2+^ from intracellular stores [35,36,37,38]. In immune cells, α7 nAChRs appear to function as metabotropic receptors rather than as ionotropic receptors [22,39,40,41]. The functional effects of dupα7 contained within metabotropic α7 nAChRs are not yet known. Considering the potential utility of α7 nAChR agonists as immunomodulatory agents [13,14,19,22,23], it is noteworthy that human peripheral blood leukocytes express more *CHRFAM7A* than *CHRNA7* [31,42,43]. We therefore investigated the mRNA expression of both α7 and dupα7 subunits in human CD4^+^ T cells and the effect of GTS-21 on Treg development. 

## 2. Results

### 2.1. mRNA Expression of α7 and Dupα7 Subunits

#### 2.1.1. Under the Resting Conditions

Figure 1 shows a scatterplot of *CHRNA7* and *CHRFAM7A* expression for each individual divided by the values closest to their respective medians to compare the magnitude of interindividual variability in *CHRNA7* and *CHRFAM7A* expression. Varied expression of both *CHRNA7* and *CHRFAM7A* were detected in all cryopreserved human peripheral blood CD4^+^ T cells examined under the resting conditions. Interindividual variation was greater for *CHRNA7* expression than for *CHRFAM7A* expression (note the log scale of the vertical axis).

#### 2.1.2. Changes during TCR-Activation

Expression of *CHRNA7* in cells cultured in RPMI medium containing 20 μg/mL IL-2 (Control) was nearly unchanged over the 7-day observation period (Figure 2A). TCR activation with Human T-activator CD3/CD28 Dynabeads did not alter *CHRNA7* expression on day 1, but the expression levels were significantly suppressed on days 4 and 7. Addition of 30 μM GTS-21 to the culture did not affect *CHRNA7* expression in the TCR-activated group over the 7-day observation period.

Under control conditions, levels of *CHRFAM7A* expression did not fluctuate over the 7-day observation period (Figure 2B). Moreover, TCR activation in the absence or presence of 30 μM GTS-21 did not affect *CHRFAM7A* expression at any time during the 7-day observation period.

### 2.2. Effects of GTS-21 on CD4^+^ T Cell Proliferation and Treg Development

#### 2.2.1. Proliferation

Numbers of CD4^+^ T cells remained nearly constant over the 7-day culture period in RPMI medium containing 20 ng/mL IL-2 (Figure 3). With TCR activation, however, CD4^+^ T cell proliferation was significantly enhanced on culture days 5 and 7, as indicated by staining with CFSE (Figure 3A) and by increasing cell number (Figure 3B). Addition of 30 μM GTS-21 caused no further enhancement of proliferation compared to TCR activation alone.

#### 2.2.2. Treg Development

Figure 4A shows flow cytometric plots for CD4^+^CD25^+^FoxP3^+^ cells (Tregs) from a representative individual in the absence and presence of 30 μM GTS-21 on day 5 of culture. The data show that GTS-21 promoted Treg development from CD4^+^ T cells.

GTS-21 promoted Treg development to varying degrees in samples from all individuals on day 5 of culture in the presence of 20 ng/mL IL-2 and 5 ng/mL transforming growth factor-β (TGF-β). The average number of Tregs was significantly higher in cultures with GTS-21 than without it (*p* = 0.008) (Figure 4B).

## 3. Discussion

### 3.1. Expression of mRNAs for α7 and Dupα7 Subunits in Resting Human CD4^+^ T Cells

Varied levels of both α7 and dupα7 subunits mRNAs were detected in cryopreserved CD4^+^ T cells from normal volunteers. However, it is not yet clear whether the large interindividual variation in α7 and dupα7 subunit expression is due to genetic disposition, immunological regulation or both. The greater interindividual variation in α7 subunit expression suggests α7 nAChRs in CD4^+^ T cells are likely more susceptible to immune stimulation in daily life. Expression of α7 and dupα7 subunit mRNAs showed no clear trends across gender, age or ethnicity.

### 3.2. Expression of α7 and Dupα7 Subunit mRNAs during TCR Activation

Changes in the mRNA expression of the α7 and dupα7 subunits during TCR activation were explored for the first time and provide valuable information about the dynamics of α7 nAChRs in human CD4^+^ T cells. Activation of TCRs on T cells up-regulates ChAT mRNA expression and ACh synthesis [8,9,10,12,44,45,46]. PHA induces ChAT mRNA expression in human mononuclear cells; induction is first evident after 48 h of exposure, and the effect persists for at least 72 h [8]. PHA-activated MOLT-3 cells, a human leukemic T cell line, secrete sufficient amounts of ACh to activate AChRs in an autocrine and paracrine manner [47]. It is therefore suggested that sustained exposure to elevated concentrations of ACh released from activated T cells attenuates *CHRNA7* expression. Indeed, the plasticity of *CHRNA7* expression has been confirmed in TCR-activated T cells [46,47,48]. The marked down-regulation of *CHRNA7* expression in TCR-activated CD4^+^ T cells after 4 days in culture observed in the present study can be attributed to negative regulation of *CHRNA7* expression mediated by sustained α7 nAChR activation induced by ACh synthesized and released from CD4^+^ T cells. This in turn suggests α7 nAChRs are involved in regulating CD4^+^ T cell function. The absence of significant variation in *CHRFAM7A* expression during TCR activation indicates dupα7 nAChRs are less involved in ACh-mediated regulation of CD4^+^ T cell function.

In contrast to the dynamic fluctuation in *CHRNA7* expression, the nearly constant *CHRFAM7A* expression in TCR-activated CD4^+^ T cells is consistent with the finding that *CHRNA7* and *CHRFAM7A* are independently regulated by their respective promoters driving their differential expression [31,42,49]. Our finding that GTS-21 did not affect *CHRFAM7A* expression in human CD4^+^ T cells differed from that in human monocyte-derived macrophages, where nicotine suppresses *CHRFAM7A* expression [26]. This suggests that regulation of *CHRFAM7A* is cell type-specific and that dupα7 nAChRs are less involved in ACh-mediated regulation of CD4^+^ T cell function.

Recent studies revealed that the α7 nAChRs have both ionotropic and metabotropic functions in neurons [36,37,50]. Moreover, most studies indicate that α7 nAChRs in macrophages and T cells function as metabotropic receptors independent of ionotropic signaling [23,39,40,41,51,52,53,54]. In oocytes injected with various combinations of dupα7 and α7 mRNAs, decreased ionotropic function of α7 nAChRs was observed with increasing dupα7/α7 composition ratios [26]. Under the present experimental conditions, α7 nAChRs in TCR-activated CD4^+^ T cells may have elevated dupα7/α7 subunit ratios after 4 days in culture and little ionotropic function. There is currently little evidence to suggest that the dupα7 subunit plays a specific physiological role in T cells, and the present findings indicate there is little if any involvement of ionotropic signaling in the regulation of CD4^+^ T cell function.

### 3.3. Effects of GTS-21 on Proliferation

In J774A.1 cells (a mouse macrophage cell line), GTS-21 mitigates LPS-induced proliferation arrest for up to 9 h [55]. In mice, proliferation of activated T cells is reportedly inhibited by stimulation of a α7 nAChR-mediated pathway [56]. In the present study, however, GTS-21 did not affect the proliferation of highly purified TCR-activated human CD4^+^ T cells. It is unclear why α7 nAChR agonists have different effects on mouse and human T cell proliferation; little information is currently available on the effects of α7 nAChR agonists, including GTS-21, on human T cell proliferation.

### 3.4. The Effect of GTS-21 on Treg Development in TCR-Activated CD4^+^ T Cells

As in our earlier study of TCR-activated CD4^+^ T cells from mice [22], GTS-21 (30 μM) promoted Treg development from human TCR-activated CD4^+^ T cells, even in the presence of appreciable levels of *CHRFAM7A* expression, thereby increasing the proportion of Tregs to about 1.8-fold over the control in 5 days. This suggests ex vivo utilization of GTS-21 could potentially help to reduce the time required to prepare sufficient numbers of Tregs for adoptive immunotherapy and reduce the associated costs. It remains possible to continue to improve Treg development through further optimization of culture conditions, including the GTS-21 concentration, beads-to-cells ratio (currently 1:1) and culture duration.

An earlier study [26] indicated that under conditions where *CHRNA7* expression is profoundly suppressed, as observed in TCR-activated CD4^+^ T cells on days 4 and 7 of the present experiment, most of the subunits that make up α7 nAChRs may be occupied by dupα7 subunits, which lack a ligand-binding site. Nevertheless, GTS-21 promoted Treg development from CD4^+^ T cell (Figure 4). Based on the observation that GTS-21 suppresses TNF-α and IL-6 secretion in LPS-stimulated macrophages from α7-KO mice, Garg and Loring (2019) reported that GTS-21 may inhibit proinflammatory cytokine production by acting at sites unrelated to α7 nAChRs [57]. However, because Mashimo et al. (2019) observed that GTS-21 promotes Treg development from wild-type CD4^+^ T cells but not in α7-KO mice [22], it seems most likely that GTS-21 promotes Treg development by acting on the α7 subunit of α7 nAChRs. However, to further confirm the involvement of α7 nAChRs in Treg development, the effects of α7 nAChR agonists other than GTS-21 should be assessed.

Metabotropic α7 nAChR function induces the release of G-proteins bound to M3-M4 loop of the channel, activating signaling cascades leading to the release of Ca^2+^ from intracellular stores [58]. The resulting increase in intracellular free Ca^2+^ ion concentration activates protein kinase C, which in turn activates PI3K/Akt signaling pathway, leading to promotion of nuclear translocation of nuclear factor erythroid 2-related factor 2 to the nucleus and overexpression of heme oxygenase-1, and finally resulting in inhibition of pro-inflammatory cytokine production, in macrophages [59,60]. Both α7 and dupα7 subunits retain G-protein binding site in their intracellular M3-M4 loop [35,37,53]. Therefore, the above process should work even if α7 subunit makes up a small fraction of the α7 nAChR. However, it remains unclear how the PI3K/Akt signaling pathway in T cells is involved in promoting Treg development.

TCR activation induces de novo IL-2 synthesis and initiates a signaling cascade that leads to activation of Janus kinase 1 (JAK1) and JAK3 [61]. IL-6 signaling is essential for optimal T cell differentiation and is transduced via JAK family proteins, culminating in STAT3 activation [62,63,64]. STAT3 is a critical positive regulator of T cell differentiation and functions in several CD4^+^ T cell subsets, including Th2 and Th17 cells and Tregs [64,65,66]. In our earlier study using TCR-activated CD4^+^ T cells from mice, GTS-21 dose-dependently enhanced IL-6 production and significantly promoted Treg development without affecting IL-2 production [22]. In the presence of IL-2 and TGF-β, GTS-21 significantly enhanced Treg development from TCR-activated human CD4^+^ T cells. These findings suggest that enhanced IL-6 production contributes to the up-regulation of Treg development by GTS-21.

In summary, results of the present study demonstrate that TCR-activation of human peripheral blood CD4^+^ T cells expressing variable levels of *CHRNA7* and *CHRFAM7A* profoundly suppressed *CHRNA7* expression with time by day 4 of the experiments, without affecting *CHRFAM7A* expression. GTS-21 did not affect the proliferation induced by TCR-activation. Despite low expression of *CHRNA7* and high expression of *CHRFAM7A*, GTS-21 promoted Treg development in human TCR-activated T cells, increasing the number of Tregs to about 1.8-fold over control in 5 days of culture. In conclusion, these findings support the ex vivo utilization of GTS-21 to reduce the time and associated costs of preparing sufficient numbers of Tregs for adoptive immunotherapy.

## 4. Materials and Methods

### 4.1. Cell Culture

Cryopreserved human peripheral blood CD4^+^ T cells from donors with various backgrounds (15 males and 3 females; 18–80 years old; Ethnicity, 4 African Americans, 4 Asians, 6 Caucasians, 3 Hispanics and 1 mixed ethnicity) (STEMCELL Technologies, Vancouver, Canada and Zen-Bio Laboratories, Durham, NC, USA) were used for this study. Vials of CD4^+^ T cells were thawed rapidly with vigorous agitation in a 37 °C water bath and washed once with RPMI 1640 (Nacalai tesque, Kyoto, Japan) supplemented with 10% fetal bovine serum Thermo Fisher Scientific (Waltham, MA, USA), 100 units/mL penicillin (Nacalai tesque), 100 μg/mL streptomycin (Nacalai tesque), 50 μM 2-mercaptoethanol (Fuji film, Tokyo, Japan) and 20 ng/mL IL-2 (BioLegend, San Diego, CA, USA) (the standard medium) at 37 °C under a humidified atmosphere with 5% CO_2_.

#### 4.1.1. CHRNA7 and CHRFAM7A Expression under the Resting Conditions

Portions of the cells from 15 samples were used for investigation of *CHRNA7* and *CHRFAM7A* expression levels under the resting conditions.

#### 4.1.2. CHRNA7 and CHRFAM7A Expression during T Cell Activation

Cells from six specimens with sufficient cell numbers were used to investigate changes in *CHRNA7* and *CHRFAM7A* expression levels during TCR-activation. The cells (1.5 × 10^5^ cells) were cultured for 7 days in triplicate in a 24-well plate containing 2 mL of the standard medium with or without 30 μM GTS-21 (Cayman Chemical Company, Ann Arbor, MI, USA). The cells were activated using Human T-activator CD3/CD28 Dynabeads (Veritas, Tokyo, Japan) at a beads-to-cell ratio of 1:1.

#### 4.1.3. Effects of GTS-21 on TCR-Activated T Cell Proliferation

To assess the effect of GTS-21 on cell proliferation, portions of the cells (2 × 10^4^ cells) from 11 samples were cultured for 7 days in duplicate in a 48-well plate containing 200 μL of the standard medium and Human T-activator CD3/CD28 Dynabeads at a beads-to-cell ratio of 1:1. After staining with 4% trypan blue, cell numbers were counted using a Countess II automated cell counter (Thermo Fisher Scientific).

For cell proliferation assay, cells were stained with 1 μM CFSE (Nacalai tesque) in PBS for 10 min and cultured under the same experimental conditions as described above for indicated times. After washing, the prepared cells were subjected to flow cytometry (CytoFLEX, Beckman Coulter, Brea, CA, USA).

#### 4.1.4. Effects of GTS-21 on Treg Development in TCR-Activated T Cells

To investigate the effect of GTS-21 on Treg development, portions of the cells (3 × 10^4^ cells) from 10 specimens were cultured for 5 days in triplicate in a 24-well plate containing 2 mL of the standard medium with 5 ng/mL TGF-β (BioLegend) and Human T-activator CD3/CD28 Dynabeads at a beads-to-cell ratio of 1:1 with or without 30 μM GTS-21.

### 4.2. Real-Time PCR

Total mRNA was extracted from human CD4^+^ T cells using Sepasol RNA II Super (Nacalai Tesque), after which cDNAs were prepared by reverse transcription using a Prime Script RT reagent Kit (Takara Bio., Shiga, Japan) in a S1000 Thermal Cycler (Bio-rad, Hercules, CA, USA). Real-time PCR analysis was conducted using TB Green Premix Ex Taq II, and predesigned primers (Takara Bio.) with a Thermal Cycler Dice Real Time System (Takara Bio.). The sequences and catalog numbers of the predesigned primers were as follows: for *CHRNA7* (HA164722), 5′-TGGCCAGATTTGGAAACCAGA-3′ (sense) and 5′-AGTGTGGAATGTGGCGTCAAAG-3 (anti-sense); for *CHRFAM7A* (HA137753), 5′-GGTTCAAGGCCAAACCGAAG-3′ (sense) and 5′-TCCTGCTGACTCAGGTGTCCA-3′ (anti-sense); and for GAPDH (HA067812), 5′-GCACCGTCAAGGCTGAGAAC-3′ (sense) and 5′-TGGTGAAGACGCCAGTGGA-3′ (anti-sense).

No amplification was observed without primers or cDNA samples. Melting curve analysis confirmed the amplicons were single at the expected melting temperature and did not form primer dimers.

*CHRNA7* and *CHRFAM7A* expression in each individual was normalized by GAPDH expression. Then to compare the magnitude of interindividual variability in *CHRNA7* and *CHRFAM7A* expression in resting CD4^+^ T cells, the levels of *CHRNA7* and *CHRFAM7A* expression in each individual were divided by the values closest to their respective medians and plotted in Figure 1.

Firstly, the expression levels of CHRNA7 and CHRFAM7A under control conditions and under TCR activation in the presence or absence of GTS-21 over time were normalized by the GAPDH expression. Next, to detect changes over time in the expression levels of *CHRNA7* (A) and *CHRFAM7A* (B) in each individual under control conditions or under TCR activation in the presence or absence of GTS-21, the ratios of GAPDH-normalized CHRNA7 and CHRFAM7A expression levels in each individual were calculated by dividing by respective GAPDH-normalized expression levels found under control conditions on day 1 when frozen/thawed samples had adapted to culture medium [67]. The ratios are shown as relative expression in Figure 2.

### 4.3. Flow Cytometry for Treg Development

To detect Tregs on day 5 of culture, cells were first washed with Hanks’ balanced salt solution supplemented with 0.1% bovine serum albumin and 0.1% NaN_3_, then stained using FITC-conjugated anti-CD4 antibody (RM4.5, Thermo Fisher Scientific) and PE-conjugated anti-CD25 antibody (PC61.5, Thermo Fisher Scientific). After subsequent fixation and permeabilization using BD Cytofix/Cytoperm solution (BD Biosciences, Franklin Lakes, NJ, USA), the cells were further stained with APC-conjugated anti-FoxP3 antibody (3G3, Thermo Fisher Scientific) and subjected to flow cytometric analysis. A gate was set on the lymphocytes using characteristic forward scatter (FSC) and side scatter (SSC) parameters. Isotype-matched FITC-, PE- and APC-conjugated mouse IgG_1_ Abs were used as controls. The acquired data was analyzed using CytExpert (Beckman Coulter). Treg development was determined by calculating the percentage of CD4^+^CD25^+^FoxP3^+^ cells cleared the gates.

### 4.4. Statistical Analysis

Data are presented as means ± S.E.M. Statistical analyses were performed using SPSS (IBM, Armonk, NY, USA). When performing parametric tests, the normality tests were performed on the data for each group. Difference of the standard deviation was assessed using F-test. Differences between two groups were evaluated using paired *t*-test, and between three or more groups using two-way analysis of variance (ANOVA) with *post-hoc* Tukey’s test, respectively. Values of *p* < 0.05 were considered as significant.

## Figures and Tables

**Figure 1 ijms-24-12257-f001:**
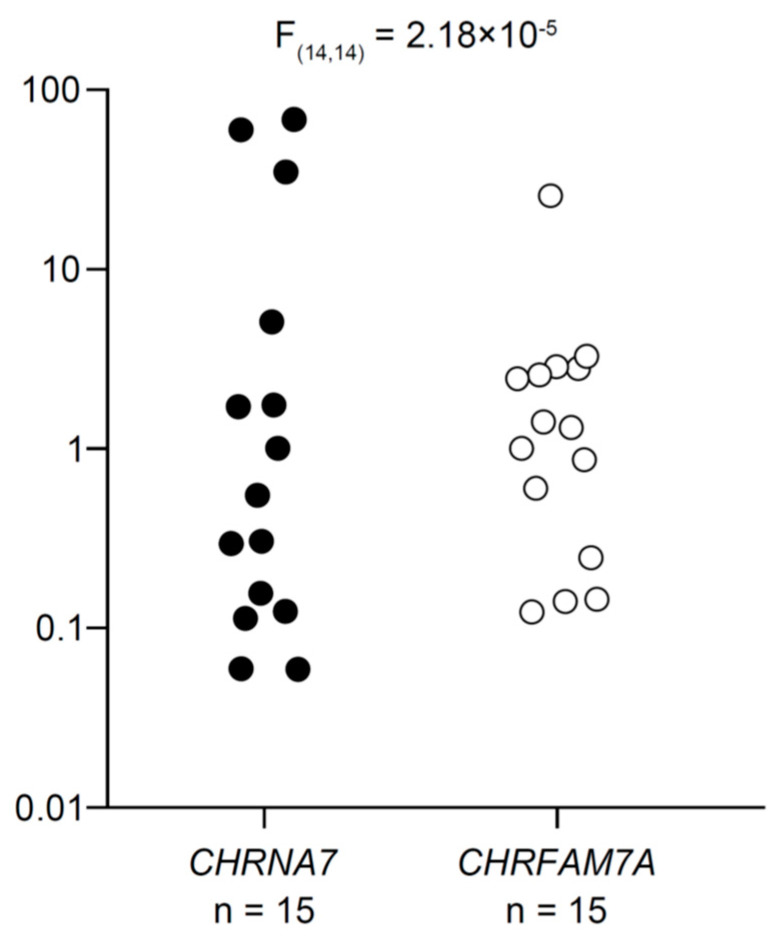
Expression of α7 and dupα7 subunit mRNAs in resting human CD4^+^ T cells. *CHRNA7* and *CHRFAM7A* (α7 and dupα7 subunit mRNA, respectively) expression levels were first normalized to *GAPDH* mRNA in each individual. Then to compare the magnitude of interindividual variability in *CHRNA7* and *CHRFAM7A* expression, levels of *CHRNA7* and *CHRFAM7A* mRNA were divided by the values closest to their respective medians and plotted on a logarithmic scale. The interindividual variability of *CHRNA7* expression was statistically greater than that of *CHRFAM7A* expression. The difference between the standard deviations was assessed with the *F*-test.

**Figure 2 ijms-24-12257-f002:**
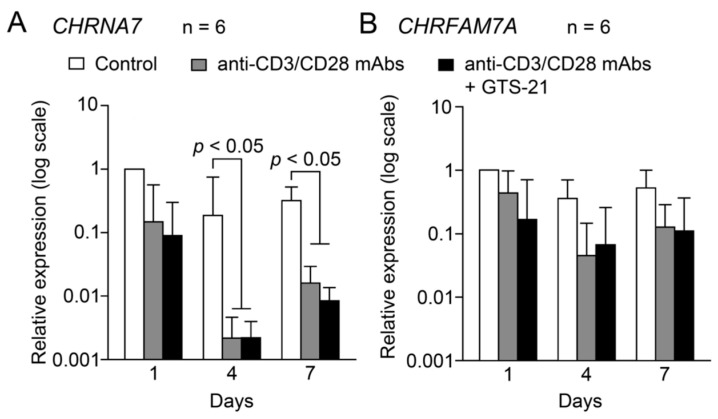
Fluctuations in mRNA expression of the α7 and dupα7 subunits during TCR activation. Human CD4^+^ T cells were cultured for up to 7 days in the standard culture medium in the presence or absence of Human T-activator CD3/CD28 Dynabeads at a beads-to-cell ratio of 1:1 with or without 30 μM GTS-21. Levels of *CHRNA7* and *CHRFAM7A* expression in cells from each individual was first normalized to *GAPDH* expression. Then to detect fluctuations over time induced by TCR activation, *CHRNA7* (**A**) and *CHRFAM7A* (**B**) mRNA levels were further divided by the respective levels in controls observed on day 1. Bars are the geomean ± S.E.M. (*n* = 6). Statistical significance was assessed with two-way ANOVA and *post-hoc* Tukey tests.

**Figure 3 ijms-24-12257-f003:**
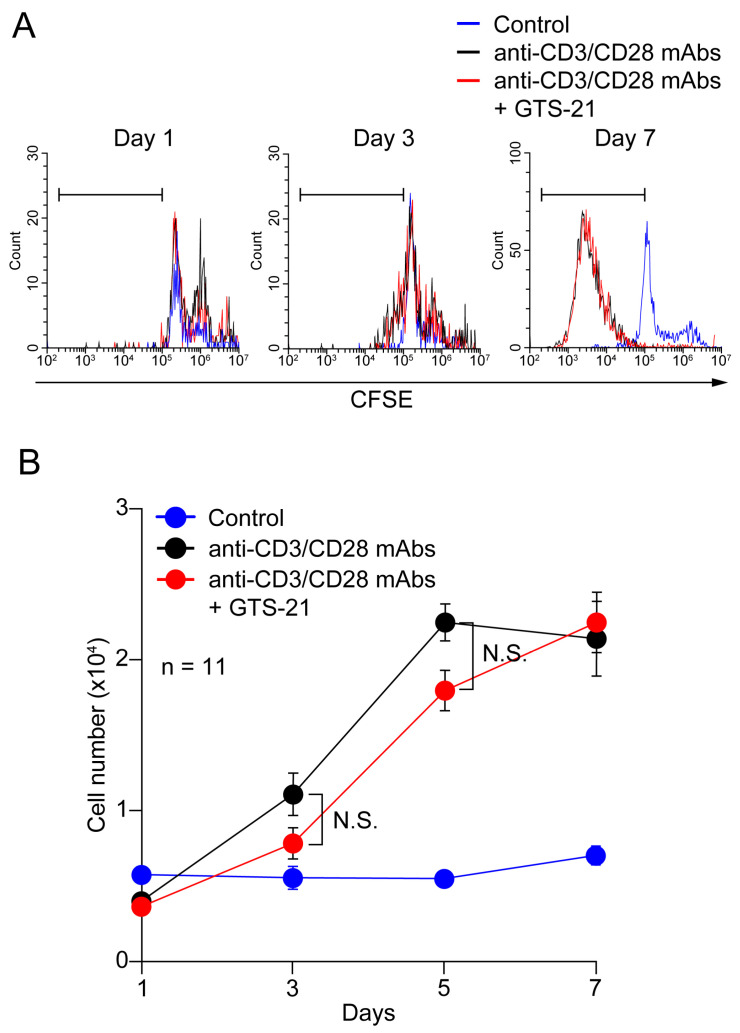
Effects of GTS-21 on CD4^+^ T cell proliferation. (**A**) Representative flow cytometric histograms for CFSE-labeled CD4^+^ T cells. Human CD4^+^ T cells were cultured for up to 7 days in the standard medium in the presence of Human T-activator CD3/CD28 Dynabeads at a beads-to-cell ratio of 1:1 with or without 30 μM GTS-21. The gates (horizontal bars) indicate proliferating cells. (**B**) Cell numbers were counted after staining with trypan blue. Data are shown as the mean ± S.E.M. (*n* = 11). N.S.; not significant (two-way ANOVA with *post-hoc* Tukey test).

**Figure 4 ijms-24-12257-f004:**
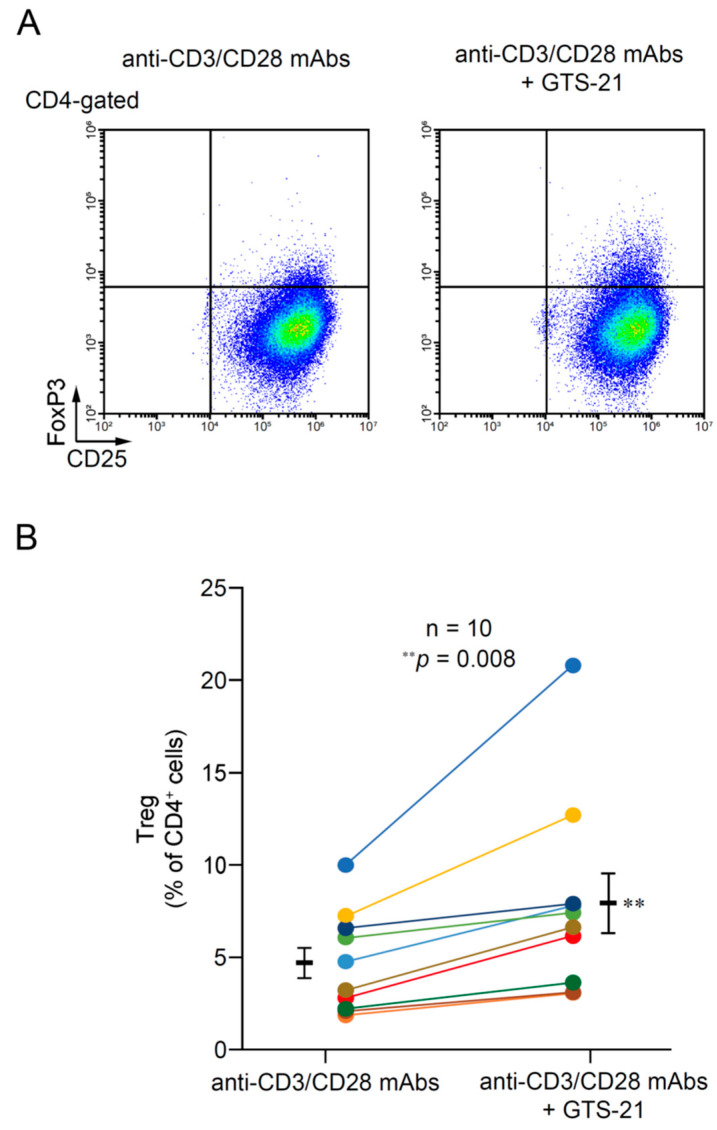
Effects of GTS-21 on Treg development. (**A**) Representative flow cytometric plots for Treg development among TCR-activated human CD4^+^ T cells in the presence or absence of 30 μM GTS-21. (**B**) GTS-21 enhanced Treg development from TCR-activated human CD4^+^ T cells on day 5 of culture. Gates were used to calculate the percentages of CD4^+^CD25^+^FoxP3^+^ cells. For comparison, the percentages of Tregs in the absence and presence of 30 μM GTS-21 in TCR-activated CD4^+^ T cells were tied by dots of the same color between the same individuals. Bars are means ± S.E.M. (*n* = 10). Statistical significance was assessed using paired *t*-tests (** *p* < 0.01).

## Data Availability

The data presented in this study are available on request from M.M. (mmashimo@dwc.doshisha.ac.jp) if and when such request is deemed appropriate and justified.

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
