# Peer review of "GTS-21 Enhances Regulatory T Cell Development from T Cell Receptor-Activated Human CD4+ T Cells Exhibiting Varied Levels of CHRNA7 and CHRFAM7A Expression"

_ijms, 2023, doi:10.3390/ijms241512257_

Round 1

Reviewer 1 Report

Mashimo et al. studied the effects of GTS-21 on the in vitro activation of CD4-positive T cells. The authors investigate the mRNA expression of CHRNA7 and CHRFAMA7A, T cell proliferation and the differentiation towards Tregs. The manuscript is well written and the experiments are well designed. However, as detailed below, some information and some essential controls are missing.

In the Material and Methods section the providers of numerous reagents are not indicated.

The mRNA expression of CHRNA7 and CHRFAMA7A is analyzed by real-time RT-PCR, which is an appropriate method. However no information is given on the controls performed along with the experiments. Did the authors control for the efficiency of the PCR? Did they control for the specificity (analysis of the size of amplicon on agarose gels and sequencing of the amplicon)? Did they include negative controls?

GAPDH was used as a housekeeping gene to normalize the results of the PCR. Did the authors control if the mRNA expression of GAPDH changes during the experiments?

The authors use parametric statistical tests. Did the authors check if the data are normally distributed, which is a prerequisite for the use of parametric tests?

Mashimo et al. studied the effects of GTS-21 on the in vitro activation of CD4-positive T cells. The authors investigate the mRNA expression of CHRNA7 and CHRFAMA7A, T cell proliferation and the differentiation towards Tregs. The manuscript is well written and the experiments are well designed. However, as detailed below, some information and some essential controls are missing.

In the Material and Methods section the providers of numerous reagents are not indicated.

The mRNA expression of CHRNA7 and CHRFAMA7A is analyzed by real-time RT-PCR, which is an appropriate method. However no information is given on the controls performed along with the experiments. Did the authors control for the efficiency of the PCR? Did they control for the specificity (analysis of the size of amplicon on agarose gels and sequencing of the amplicon)? Did they include negative controls?

GAPDH was used as a housekeeping gene to normalize the results of the PCR. Did the authors control if the mRNA expression of GAPDH changes during the experiments?

The authors use parametric statistical tests. Did the authors check if the data are normally distributed, which is a prerequisite for the use of parametric tests?

Mashimo et al. studied the effects of GTS-21 on the in vitro activation of CD4-positive T cells. The authors investigate the mRNA expression of CHRNA7 and CHRFAMA7A, T cell proliferation and the differentiation towards Tregs. The manuscript is well written and the experiments are well designed. However, as detailed below, some information and some essential controls are missing.

In the Material and Methods section the providers of numerous reagents are not indicated.

The mRNA expression of CHRNA7 and CHRFAMA7A is analyzed by real-time RT-PCR, which is an appropriate method. However no information is given on the controls performed along with the experiments. Did the authors control for the efficiency of the PCR? Did they control for the specificity (analysis of the size of amplicon on agarose gels and sequencing of the amplicon)? Did they include negative controls?

GAPDH was used as a housekeeping gene to normalize the results of the PCR. Did the authors control if the mRNA expression of GAPDH changes during the experiments?

The authors use parametric statistical tests. Did the authors check if the data are normally distributed, which is a prerequisite for the use of parametric tests?

Mashimo et al. studied the effects of GTS-21 on the in vitro activation of CD4-positive T cells. The authors investigate the mRNA expression of CHRNA7 and CHRFAMA7A, T cell proliferation and the differentiation towards Tregs. The manuscript is well written and the experiments are well designed. However, as detailed below, some information and some essential controls are missing.

In the Material and Methods section the providers of numerous reagents are not indicated.

The mRNA expression of CHRNA7 and CHRFAMA7A is analyzed by real-time RT-PCR, which is an appropriate method. However no information is given on the controls performed along with the experiments. Did the authors control for the efficiency of the PCR? Did they control for the specificity (analysis of the size of amplicon on agarose gels and sequencing of the amplicon)? Did they include negative controls?

GAPDH was used as a housekeeping gene to normalize the results of the PCR. Did the authors control if the mRNA expression of GAPDH changes during the experiments?

The authors use parametric statistical tests. Did the authors check if the data are normally distributed, which is a prerequisite for the use of parametric tests?

Mashimo et al. studied the effects of GTS-21 on the in vitro activation of CD4-positive T cells. The authors investigate the mRNA expression of CHRNA7 and CHRFAMA7A, T cell proliferation and the differentiation towards Tregs. The manuscript is well written and the experiments are well designed. However, as detailed below, some information and some essential controls are missing.

In the Material and Methods section the providers of numerous reagents are not indicated.

The mRNA expression of CHRNA7 and CHRFAMA7A is analyzed by real-time RT-PCR, which is an appropriate method. However no information is given on the controls performed along with the experiments. Did the authors control for the efficiency of the PCR? Did they control for the specificity (analysis of the size of amplicon on agarose gels and sequencing of the amplicon)? Did they include negative controls?

GAPDH was used as a housekeeping gene to normalize the results of the PCR. Did the authors control if the mRNA expression of GAPDH changes during the experiments?

The authors use parametric statistical tests. Did the authors check if the data are normally distributed, which is a prerequisite for the use of parametric tests?

Mashimo et al. studied the effects of GTS-21 on the in vitro activation of CD4-positive T cells. The authors investigate the mRNA expression of CHRNA7 and CHRFAMA7A, T cell proliferation and the differentiation towards Tregs. The manuscript is well written and the experiments are well designed. However, as detailed below, some information and some essential controls are missing.

In the Material and Methods section the providers of numerous reagents are not indicated.

The mRNA expression of CHRNA7 and CHRFAMA7A is analyzed by real-time RT-PCR, which is an appropriate method. However no information is given on the controls performed along with the experiments. Did the authors control for the efficiency of the PCR? Did they control for the specificity (analysis of the size of amplicon on agarose gels and sequencing of the amplicon)? Did they include negative controls?

GAPDH was used as a housekeeping gene to normalize the results of the PCR. Did the authors control if the mRNA expression of GAPDH changes during the experiments?

The authors use parametric statistical tests. Did the authors check if the data are normally distributed, which is a prerequisite for the use of parametric tests?

Mashimo et al. studied the effects of GTS-21 on the in vitro activation of CD4-positive T cells. The authors investigate the mRNA expression of CHRNA7 and CHRFAMA7A, T cell proliferation and the differentiation towards Tregs. The manuscript is well written and the experiments are well designed. However, as detailed below, some information and some essential controls are missing.

In the Material and Methods section the providers of numerous reagents are not indicated.

The mRNA expression of CHRNA7 and CHRFAMA7A is analyzed by real-time RT-PCR, which is an appropriate method. However no information is given on the controls performed along with the experiments. Did the authors control for the efficiency of the PCR? Did they control for the specificity (analysis of the size of amplicon on agarose gels and sequencing of the amplicon)? Did they include negative controls?

GAPDH was used as a housekeeping gene to normalize the results of the PCR. Did the authors control if the mRNA expression of GAPDH changes during the experiments?

The authors use parametric statistical tests. Did the authors check if the data are normally distributed, which is a prerequisite for the use of parametric tests?

Mashimo et al. studied the effects of GTS-21 on the in vitro activation of CD4-positive T cells. The authors investigate the mRNA expression of CHRNA7 and CHRFAMA7A, T cell proliferation and the differentiation towards Tregs. The manuscript is well written and the experiments are well designed. However, as detailed below, some information and some essential controls are missing.

In the Material and Methods section the providers of numerous reagents are not indicated.

The mRNA expression of CHRNA7 and CHRFAMA7A is analyzed by real-time RT-PCR, which is an appropriate method. However no information is given on the controls performed along with the experiments. Did the authors control for the efficiency of the PCR? Did they control for the specificity (analysis of the size of amplicon on agarose gels and sequencing of the amplicon)? Did they include negative controls?

GAPDH was used as a housekeeping gene to normalize the results of the PCR. Did the authors control if the mRNA expression of GAPDH changes during the experiments?

The authors use parametric statistical tests. Did the authors check if the data are normally distributed, which is a prerequisite for the use of parametric tests?

Mashimo et al. studied the effects of GTS-21 on the in vitro activation of CD4-positive T cells. The authors investigate the mRNA expression of CHRNA7 and CHRFAMA7A, T cell proliferation and the differentiation towards Tregs. The manuscript is well written and the experiments are well designed. However, as detailed below, some information and some essential controls are missing.

In the Material and Methods section the providers of numerous reagents are not indicated.

The mRNA expression of CHRNA7 and CHRFAMA7A is analyzed by real-time RT-PCR, which is an appropriate method. However no information is given on the controls performed along with the experiments. Did the authors control for the efficiency of the PCR? Did they control for the specificity (analysis of the size of amplicon on agarose gels and sequencing of the amplicon)? Did they include negative controls?

GAPDH was used as a housekeeping gene to normalize the results of the PCR. Did the authors control if the mRNA expression of GAPDH changes during the experiments?

The authors use parametric statistical tests. Did the authors check if the data are normally distributed, which is a prerequisite for the use of parametric tests?

Author Response

Response to Reviewer 1 Comments

First of all, we would like to thank the reviewer for these very insightful comments.

Point 1: In the Materials and Methods section the providers of numerous reagents are not indicated.

Response 1: In accordance with the reviewer’s criticism, we have inserted the names of the reagent suppliers of reagents in the Materials and Methods section on pages 8-9.

Point 2: The mRNA expression of CHRNA7 and CHRFAMA7A is analyzed by real-time RT-PCR, which is an appropriate method.  However, no information is given on the controls performed along with the experiments. Did the authors control for the specificity of the PCR? Did they control for the specificity (analysis of the size of amplicon on agarose gels and sequencing of the amplicon)? Did they include negative controls?

Response 2: Regarding the specificity of PCR, melting curve analysis confirmed that the amplicons were single at the expected melting temperatures and did not form primer dimers. In this way we confirmed the specificity of the primers. 

To refer to the specificity of primers, on page 9, lines 328-329, we have inserted the sentence “Melting curve analysis confirmed the amplicons were single at the expected melting temperature and did not form primer dimers.”

We did not run negative controls.

Point 3: GAPDH was used as a housekeeping gene to normalize the results of the PCR. Did the authors control if the mRNA expression of GAPDH changes during the experiments?

Response 3: We thank the reviewer for reminding us of the reliability issues of GAPDH as housekeeping gene. In reference to the precedent studies on the expression of CHRNA7 and CHRFAM7A in human leukocytes (Costantini et al, 2015 (Ref 31)), we determined CHRNA7 and CHRFAM7A expression using GAPDH as housekeeping gene.

Expression of GAPDH levels was stable throughout the experimental period of 1-5 days in control and TCR-activated groups with or without GTS-21. However, recent study by Roy et al (2020) has raised concerns about the reliability of GAPDH as a housekeeping gene. Therefore, in future projects, we would like to consider employing other housekeeping gene such as ACTB.

Roy, J.G., McElhaney, J.E. & Verschoor, C.P. Reliable reference genes for the quantification of mRNA in human T-cells and PBMCs stimulated with live influenza virus. BMC Immunol 21, 4 (2020). https://doi.org/10.1186/s12865-020-0334-8.

Point 4: The authors use parametric statistical tests. Did the authors check if the data are normally distributed, which is a prerequisite for the use of parametric tests?

Response 4: We performed statistical analyses using SPSS (IBM, Armonk, NY, USA), as described in the section 4.4. on page 9, lines 344-346. A normality test of the data for each group was always performed before running the parametric test. To explain this, we have inserted the sentence “When performing parametric tests, the normality tests were performed on the data for each group.”

Reviewer 2 Report

In this study, the results indicate that the selective α7 nAChR agonist GTS-21 enhances regulatory T cell development from T cell receptor-activated human CD4+ T cells exhibiting varied levels of CHRNA7 and CHRFAM7A expression. Despite the study is interesting I have several concerns:

1) In figure 1, authors should include statistical results in the graph.

2) Authors should perform some cell proliferation assays such as flow cytometry or immunofluorescence to assess the expression of the proliferation marker PCNA. This is essential to confirm the result on CD4+ T cell proliferation.  

3) Concerning Treg development, authors should perform immunofluorescence to show the development of Treg cells upon treatment with GTS-21.

Author Response

Response to Reviewer 2 Comments

First of all, we would like to thank the reviewer for these very insightful comments.

Point 1: In figure 1, authors should include statistical results in the graph.

Response 1: In accordance with the reviewer’s suggestion, we have inserted the statistical result (F value) at the top of the revised Figure 1 and deleted from the legend.  

Point 2: Authors should perform some cell proliferation assays such as flow cytometry or immunofluorescence to assess the expression of the proliferation marker PCNA. This is essential to confirm the results on CD4+ T cell proliferation.

Response 2: We would like to point out that since we used isolated human CD4+ T cells in the present study, cell number measurements should be sufficient to detect proliferation induced by different treatments.  However, we also confirmed human CD4+ T cell proliferation using CFSE.  Therefore, we modified the Figure 3 by inserting the graph of CFSE staining as Figure 3A and left the previous Figure 3 as Figure 3B.

Accordingly,

  • on page 4, lines 134-135, we have inserted the phrase “as indicated by staining with CFSE (Figure 3A) and by increasing cell number (Figure 3B)” to describe the results on CFSE assay.
  • on page 5, lines 158-161, we have inserted two sentences “(A) Representative flow cytometric histograms for CFSE-labeled CD4+ Human CD4+ T cells were cultured for up to 7 days in the standard medium in the presence of Human T-activator CD3/CD28 Dynabeads at a beads-to-cell ratio of 1:1 with or without 30 mM GTS-21. The gates indicate proliferating cells.”
  • on page 8, lines 299-303, we have inserted the descriptions on the procedure of CFSE assay.

Point 3: Concerning Treg development, authors should perform immunofluorescence to show the development of Treg cells upon treatment with GTS-21.

Response 3: We believe the procedure used for Treg detection in the present study is one of the widely accepted and standard procedures.  For clarity, on page 9, lines 340-342, we have inserted the sentence “Treg development was determined by calculating the percentage of CD4+CD25+FoxP3+ cells cleared the gates.”

Round 2

Reviewer 1 Report

Mashimo et al. submitted a revised version of their manuscript entitled “GTS-21 enhances regulatory T cell development from T cell receptor-activated human CD4+ T cells exhibiting varied levels of CHRNA7 and CHRFAM7A expression”.

The revised manuscript is improved but there are still critical points regarding essential controls needed for the real-time RT-PCR experiments. In my previous review, I asked the following questions:

-          Did the authors control for the efficiency of the PCR? Did they control for the specificity (analysis of the size of amplicon on agarose gels and sequencing of the amplicon)? Did they include negative controls?

-          GAPDH was used as a housekeeping gene to normalize the results of the PCR. Did the authors control if the mRNA expression of GAPDH changes during the experiments?

These questions were not appropriately answered by the authors. Proper controls are, however, essential to judge the quality of the PCR data. The authors newly included the information that the melting curve was OK, which only tells us most probably a single amplicon was produced. As the expression levels of CHRNA7 and CHRFAM7A are of high importance for this study, proper controls are mandatory.

Mashimo et al. submitted a revised version of their manuscript entitled “GTS-21 enhances regulatory T cell development from T cell receptor-activated human CD4+ T cells exhibiting varied levels of CHRNA7 and CHRFAM7A expression”.

The revised manuscript is improved but there are still critical points regarding essential controls needed for the real-time RT-PCR experiments. In my previous review, I asked the following questions:

-          Did the authors control for the efficiency of the PCR? Did they control for the specificity (analysis of the size of amplicon on agarose gels and sequencing of the amplicon)? Did they include negative controls?

-          GAPDH was used as a housekeeping gene to normalize the results of the PCR. Did the authors control if the mRNA expression of GAPDH changes during the experiments?

These controls are, however, essential to judge the quality of the PCR data. The authors newly included the information that the melting curve was OK, which only tells us most probably a single amplicon was produced. As the expression levels of CHRNA7 and CHRFAM7A are of high importance for this study, proper controls are mandatory.

Mashimo et al. submitted a revised version of their manuscript entitled “GTS-21 enhances regulatory T cell development from T cell receptor-activated human CD4+ T cells exhibiting varied levels of CHRNA7 and CHRFAM7A expression”.

The revised manuscript is improved but there are still critical points regarding essential controls needed for the real-time RT-PCR experiments. In my previous review, I asked the following questions:

-          Did the authors control for the efficiency of the PCR? Did they control for the specificity (analysis of the size of amplicon on agarose gels and sequencing of the amplicon)? Did they include negative controls?

-          GAPDH was used as a housekeeping gene to normalize the results of the PCR. Did the authors control if the mRNA expression of GAPDH changes during the experiments?

These controls are, however, essential to judge the quality of the PCR data. The authors newly included the information that the melting curve was OK, which only tells us most probably a single amplicon was produced. As the expression levels of CHRNA7 and CHRFAM7A are of high importance for this study, proper controls are mandatory.

Mashimo et al. submitted a revised version of their manuscript entitled “GTS-21 enhances regulatory T cell development from T cell receptor-activated human CD4+ T cells exhibiting varied levels of CHRNA7 and CHRFAM7A expression”.

The revised manuscript is improved but there are still critical points regarding essential controls needed for the real-time RT-PCR experiments. In my previous review, I asked the following questions:

-          Did the authors control for the efficiency of the PCR? Did they control for the specificity (analysis of the size of amplicon on agarose gels and sequencing of the amplicon)? Did they include negative controls?

-          GAPDH was used as a housekeeping gene to normalize the results of the PCR. Did the authors control if the mRNA expression of GAPDH changes during the experiments?

These controls are, however, essential to judge the quality of the PCR data. The authors newly included the information that the melting curve was OK, which only tells us most probably a single amplicon was produced. As the expression levels of CHRNA7 and CHRFAM7A are of high importance for this study, proper controls are mandatory.

Author Response

July 21, 2023

Mr. Kotchanat Srisangchun

Assistant Editor

IJMS

Re: Manuscript ID: ijms-2475964 -Revision-2

Dear Mr. Srisangchun,

Uploaded please find our revised manuscript entitled “GTS-21 enhances regulatory T cell development from T cell receptor-activated human CD4+ T cells exhibiting varied levels of CHRNA7 and CHRFAM7A expression” (ijms-2475964-R2-230719-KK) by Mashimo et al.  I would like to thank you and the reviewer for comments. 

To address the issue raised by the reviewer regarding the quality of our real-time PCR, we performed additional experiments on negative controls proving the reliability of our PCR procedure. In addition, we have rewritten the sentences in the text regarding the detection of the expression levels of CHRNA7 (A) and CHRFAM7A (B) shown in Figure 2. For further details, please see the responses to the reviewer comments. All of the changes made were highlighted in red in the revised manuscript.

It is our hope that the revised manuscript is acceptable now for publication in the International Journal of Molecular Sciences.

Sincerely yours,

Koichiro Kawashima, Ph.D.

Visiting Professor

Kitasato University School of Pharmaceutical Sciences

Tokyo 108-8641, Japan

Tel/Fax: +45-983-5964

E-mail: [email protected]; [email protected]

Reviewer 2 Report

Authors addressed all my concerns. 

Author Response

(The authors gave the same response as above.)

Round 3

Reviewer 1 Report

The authors approriately revised their manuscript.